# Systemic nicotinamide mononucleotide administration to mitigate post-cardiac arrest brain injury in mice

Daiki Kaito[1], Tomoyoshi Tamura[1,*], Sayuri Suzuki[1], Ryutaro Onishi[1,2], Kenji F. Tanaka[2], Jun Yoshino[3,4], Tadashi Matsuoka[1], Katsuya Maeshima[1], Junichi Sasaki[1], Koichiro Homma[1,*]

1 Department of Emergency and Critical Care Medicine, Keio University School of Medicine, Shinjuku, Tokyo, Japan, 2 Division of Brain Sciences, Institute for Advanced Medical Research, Keio University School of Medicine, Shinjuku, Tokyo, Japan, 3 Division of Nephrology, Department of Internal Medicine, Faculty of Medicine, Shimane University, Izumo, Shimane, Japan, 4 The Center for Integrated Kidney Research and Advance (IKRA), Faculty of Medicine, Shimane University, Izumo, Shimane, Japan

☯ These authors contributed equally to this work.
* tomoyoshitamura@keio.jp (TT); homma@keio.jp (KH)

## Abstract

Post-cardiac arrest brain injury (PCABI) is the leading cause of death and disability following cardiac arrest (CA). Nicotinamide adenine dinucleotide ($NAD^+$) depletion after CA contributes to neuronal injury, while nicotinamide mononucleotide (NMN) replenishes $NAD^+$ and may provide neuroprotection via sirtuin activation. This study aimed to investigate the effects of systemic NMN administration on neurological function, survival, and sirtuin-3 (SIRT3) levels in the brain post-CA. In adult male mice (C57BL/6NCrSlc, 10–15 weeks old), 10-min CA was induced by intravenous potassium chloride injection followed by cardiopulmonary resuscitation. NMN (60 mg/kg body weight) or normal saline (control) was randomly administered by intraperitoneal injection immediately after the return of spontaneous circulation (ROSC) and 24 and 48 h post-CA. Brain $NAD^+$ and adenosine triphosphate (ATP) levels, neurological function score (NFS), survival, histological neuronal injury, and brain gene expression and protein levels were measured. Brain $NAD^+$ levels decreased at 2 h post-ROSC and NMN significantly increased brain $NAD^+$ and ATP levels. At 48 h post-CA, surviving mice in the NMN group exhibited significantly higher NFS (control: 8 [IQR: 4–12] vs. NMN: 12 [IQR: 9–12], p = 0.03) and less severe hippocampal neuronal injury compared with controls. Moreover, the NMN group showed significantly higher 7-day survival rate (control: 22.2% [4/18] vs. NMN: 61.1% [11/18], p = 0.03) and brain SIRT3 levels (control: 17.7 ± 3.6 vs. NMN: 34.5 ± 4.4 pg/mg protein, p = 0.01). In conclusion, systemic NMN administration after ROSC attenuates PCABI. The increased brain ATP levels and SIRT3 upregulation may suggest the usefulness of NMN for improving mitochondrial function and contributing to neuroprotection. $NAD^+$ supplementation with NMN is a promising therapeutic approach against PCABI.

**Data availability statement:** All relevant data are within the paper and its Supporting Information files.

**Funding:** This research was supported by a Grants-in-Aid for Scientific Research from the Japan Society for the Promotion of Science (KAKENHI, grant number 23K27703 to K.H.; 22K16632 to T.T.; 20K23382, 24K02506 to J.Y. https://www.jsps.go.jp/) and the grant from the Astellas Foundation for Research on Metabolic Disorders (https://www.astellas-foundation.or.jp/). The funding sources had no role in study design, data collection, analysis, interpretation, writing of the report, or decision to submit the manuscript for publication.

**Competing interests:** I have read the journal's policy and the authors of this manuscript have the following competing interests: [J.Y. is listed as an inventor on patent applications related to NMN and adiponectin (US20180228824, JP2018131418A). The other authors declare no conflict of interest.] This does not alter our adherence to PLOS ONE policies on sharing data and materials.

## Introduction

Post-cardiac arrest brain injury (PCABI) remains the leading cause of death after cardiac arrest (CA) [1]. The mainstay of treatment against PCABI is limited to supportive therapies, and definitive treatment modalities are lacking. Therefore, novel neuroprotective therapies are required to mitigate PCABI [1].

Ischemia-reperfusion injury triggers PCABI involving mitochondrial dysfunction, release of reactive oxygen species (ROS), and inflammation [1]. Nicotinamide adenine dinucleotide ($NAD^+$) depletion after ischemia-reperfusion, including post-CA, contributes to neuronal injury [2,3]. During ischemia-reperfusion, ROS overactivate poly(ADP-ribose) polymerase (PARP), which depletes $NAD^+$ to repair DNA damage [2]. This depletion limits the activity of other $NAD^+$-dependent enzymes, especially sirtuins that affect mitochondrial function, inflammation, and cell survival [2,4]. Therefore, sirtuin activation through $NAD^+$ supplementation is an attractive acute therapeutic target after ischemia-reperfusion [4]. Previous studies have shown that $NAD^+$ supplementation with niacin or exogenous $NAD^+$ mitigates PCABI [3,5]. However, the clinical translation of such $NAD^+$ supplementation is hindered by niacin-induced flushing and the poor cellular uptake of $NAD^+$ itself [6,7].

Nicotinamide mononucleotide (NMN), a direct precursor of $NAD^+$, can supplement $NAD^+$ and has been investigated as a treatment for ischemia-reperfusion injuries [8]. In a rodent model of hemorrhagic shock, NMN increased tissue $NAD^+$ and adenosine triphosphate (ATP) levels, reduced serum interleukin-6 (IL-6) levels, and improved short-term survival [9]. Furthermore, in a murine model of cerebral ischemia, NMN alleviated post-ischemic reduction in brain $NAD^+$ levels and attenuated neuronal injury via mechanisms dependent on sirtuin-3 (SIRT3), a key regulator of mitochondrial function and inflammation [10–13]. Finally, preliminary human trials have demonstrated the safety of NMN, suggesting its potential for clinical application [14,15]. Collectively, these findings indicate that NMN may have the potential to mitigate PCABI.

This study aimed to investigate the effects of NMN on neurological outcomes after CA in a murine model. We hypothesized that NMN would increase $NAD^+$ and ATP levels in the brain, upregulate SIRT3, and improve neurological function and survival after CA. Based on previous studies demonstrating that intraperitoneal NMN administration induces a rapid and sustained increase in $NAD^+$ levels, the intraperitoneal route was selected over intravenous administration in this study [10–12].

## Materials and methods

### Study design

This was a murine comparative study in which mice were randomly assigned to either the NMN or control group using a computer-generated randomization list prepared by an investigator (D.K.). This study was conducted in strict accordance with the recommendations in the Guidelines for Proper Conduct of Animal Experiments issued by the Science Council of Japan. The protocol was approved by the Keio University Institutional Animal Care and Use Committee (approval number: A2022-154), and all experiments were performed in compliance with the Institutional Guidelines on

Animal Experimentation at Keio University. Additionally, this study adhered to the Animal Research: Reporting of In Vivo Experiments guidelines [16].

We conducted 4 experiments: brain NAD$^+$ and ATP assays, neurological function and survival analysis, brain histological analysis, and brain RNA and enzyme-linked immunosorbent assay (ELISA) analyses (Fig 1). The primary outcome measure of the neurological function and survival analysis was the neurological function score (NFS) at 48 h post-CA among surviving mice, while the secondary outcomes were the NFS 24 h post-CA among surviving mice and 7-day survival.

The target sample sizes for the outcomes were calculated to provide 80% power to detect differences based on previous reports and our preliminary experiments, with a two-sided alpha level of 0.05. For the brain NAD$^+$ assay, 32 mice were required. Particularly, a sample size of 14 mice was estimated to detect a mean difference of 20 pmol/mg tissue (standard deviation [SD]: 12 pmol/mg tissue) in brain NAD$^+$ levels between groups at 15 min after the return of spontaneous circulation (ROSC) (control: 160 vs. NMN: 180 pmol/mg tissue) [3,11,17]. A sample size of 14 mice was estimated to detect a mean difference of 60 pmol/mg tissue (SD: 35 pmol/mg tissue) in brain NAD$^+$ levels between groups at 2 h post-ROSC (control: 140 vs. NMN 200 pmol/mg tissue) [3,10,11,17]. Finally, a sample size of 4 sham-operated mice was estimated to detect a mean difference of 50 pmol/mg tissue (SD: 22 pmol/mg tissue) in brain NAD$^+$ levels between groups (sham: 170 vs. control [2 h post-ROSC]: 120 pmol/mg tissue) [3,10,11,17]. For the brain ATP assay, a sample size of 10 mice was estimated to detect a mean difference of 150 nmol/g tissue (SD: 70 nmol/g tissue) in brain ATP levels between groups at 2 h post-ROSC (control: 250 vs. NMN: 400 nmol/g tissue) [5,11,18–20].

For the neurological function and survival analysis, 36 mice were required (18 mice per group). Particularly, a sample size of 20 mice was estimated to detect a mean difference of 4 (SD: 3) in NFS between groups at 48 h post-CA among surviving mice (control: 7 vs. NMN: 11) [3,10,21], and a sample size of 36 mice was estimated to detect an absolute 40% difference (control: 20% vs. NMN: 60%) in the survival rate [5,9,22]. For the histological analysis, 20 mice were required. A

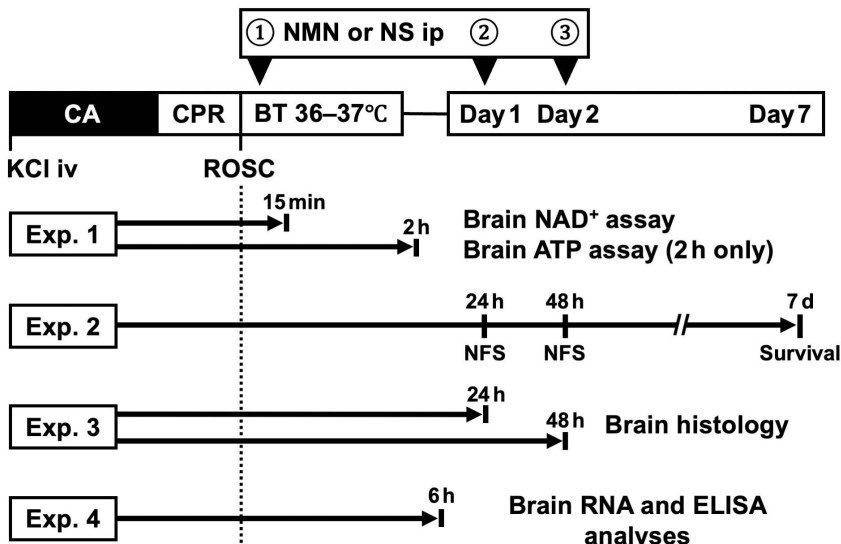

**Fig 1. Schematic summary of the experimental protocols.** Mice are assigned to the NMN group (NMN 60 mg/kg) or the control group (equivalent volume of NS). NMN or NS is administered at 0, 24, and 48 h after cardiac arrest. The following 4 experiments are conducted: experiment 1: brain NAD$^+$ and ATP assays; experiment 2: neurological function and survival analysis; experiment 3: histological analysis of hippocampi; and experiment 4: brain RNA and ELISA analyses. ATP, adenosine triphosphate; BT, body temperature; CA, cardiac arrest; CPR, cardiopulmonary resuscitation; ELISA, enzyme-linked immunosorbent assay; Exp, experiment; KCl, potassium chloride; NAD$^+$, nicotinamide adenine dinucleotide; NFS, neurological function score; NMN, nicotinamide mononucleotide; NS, normal saline; RNA, ribonucleic acid; ROSC, return of spontaneous circulation.

sample size of 10 mice was estimated to detect a mean difference of 7% (SD: 3%) in Fluoro-Jade C (FJC)-positive areas between groups at 48 h post-ROSC (control: 10% vs. NMN: 3%) [3,10,23]. To follow the time course, the same number of mice was used at 24 h post-ROSC.

For the brain RNA analysis, 6 mice were required. We estimated a sample size of 6 mice to detect a mean difference of 0.5 (SD: 0.1) in relative *Sirt3* expression between groups (control: 0.5 vs. NMN: 1.0) [5,20,24]. A sample size of 6 mice was also estimated to detect a mean difference of 0.5 (SD: 0.1) in relative *Il6* expression between groups (control: 1.0 vs. NMN: 0.5) [23,25]. For the brain ELISA analysis, 12 mice were required. We estimated a sample size of 12 mice to detect a mean difference of 10 pg/mg protein (SD: 10 pg/mg protein) in SIRT3 protein levels between groups (control: 20 vs. NMN: 40 pg/mg protein) [5,20,24,26]. A sample size of 12 mice was also estimated to detect a mean difference of 1.0 pg/mg protein (SD: 0.5 pg/mg protein) in IL-6 protein levels between groups (control: 2.0 vs. NMN: 1.0 pg/mg protein) [23,25,27].

Considering the possibility of some mice not achieving ROSC, additional mice beyond the required sample size were included. All surgeries were performed under isoflurane anesthesia combined with local lidocaine. For sample collection, mice were sacrificed by cervical dislocation under anesthesia induced by the intraperitoneal administration of midazolam, medetomidine, and butorphanol.

This study was single-blinded. The investigators who performed surgery, cardiopulmonary resuscitation (CPR), and NMN or normal saline (NS) administration were aware of the treatment assignment. However, assessments of neurological function and histologic neuronal injury were performed by investigators blinded to treatment assignment. This study was not registered to any protocol repository.

## Animals

Adult male mice (C57BL/6NCrSlc, 10–15 weeks old) were purchased from Japan SLC (Hamamatsu, Japan) and were allowed free access to water and standard rodent food in a temperature-controlled room (23℃) with a 12-h light-dark cycle. All mice were acclimatized for at least 1 week before the experiments and kept at the same height in the same rack, regardless of the assigned group.

## Mouse model of CA and resuscitation

We used an established murine CA model [28]. Briefly, the mice were anesthetized using isoflurane, intubated, and mechanically ventilated (mini-vent; Harvard Apparatus, Holliston, MA, USA). $FiO_2$ was set at 1.0, tidal volume at 8 μL/g body weight, and frequency at 150 breaths/min. The right femoral artery and vein were catheterized surgically, and body temperature, arterial blood pressure, and heart rate were monitored during the experiments (PowerLab System; LabChart; AD Instruments, Bella Vista, NSW, Australia). Cardiac arrest was induced by the intravenous administration of potassium chloride at a dose of 80 μg/g body weight and was maintained for 10 min. Cardiopulmonary resuscitation was started with chest compressions manually at a frequency greater than 300 times/min, along with mechanical ventilation and continuous intravenous injection of 0.024 μg/g/min epinephrine. We defined ROSC as a mean arterial pressure (MAP) of > 40 mmHg that lasted at least 1 min. If ROSC was not restored after 4 min of CPR, resuscitation was ceased. After ROSC, the body temperature was controlled to 36–37℃ for 1 h. Epinephrine infusion was tapered and discontinued to maintain a MAP of 70 mmHg. All experiments were conducted in a dedicated laboratory space.

## NMN administration protocol

NMN at a dose of 60 mg/kg dissolved in 200 μL NS was immediately administered post-ROSC via intraperitoneal injection as described in a previous murine model of cerebral ischemia [10]. The control group received the same amount of NS via intraperitoneal injection. The same administration of NMN or NS was repeated 24 and 48 h post-CA. NMN was purchased from Oriental Yeast Co. Ltd. (Tokyo, Japan).

## Brain NAD+ assay

Mitochondrial injury occurs rapidly after ROSC [29], making the timely transport of NMN across the blood–brain barrier crucial for preventing and treating PCABI. A previous study reported that NMN rapidly increased brain NAD+ levels [11]. Therefore, we initially measured NAD+ concentrations in the brain following post-ROSC systemic administration of NMN via intraperitoneal injection. At 15 min and 2 h after ROSC, brain tissues that included the cortex and hippocampus (≤ 20 mg) were harvested and snap frozen with liquid nitrogen. Sham-operated mice underwent the same procedure, including tracheal intubation and arteriovenous cannulation, but CA was not induced. Brain tissues were homogenized, and brain NAD+ concentrations were measured using the NAD+ quantitation kit (EnzyChrom NAD/NADH Assay Kit [E2ND-100]; BioAssay Systems, Hayward, CA, USA), a highly specific kit for NAD+/NADH, according to the manufacturer's protocol.

## Brain ATP assay

NMN treatment can support mitochondrial function as evidenced by higher brain tissue ATP levels [4]. Thus, ATP levels in brain tissues, including the cortex and hippocampus, at 2 h post-ROSC were measured using a luciferase ATP level assay kit (AMERIC-ATP(T) kit; Applied Medical Enzyme Research Institute Corporation, Tokushima, Japan) according to the manufacturer's protocol. The brain tissues were homogenized with phenol-containing extraction reagent and centrifuged at 10,000 × g for 10 min, and the supernatants were collected. The supernatants were diluted 100-fold with ultrapure sterile water, and 10 µL of these diluted supernatants were mixed with 90 µL luciferase reagent. Then, bioluminescence was measured using a luminometer. Brain ATP levels (nmol/g tissue) in each sample were calculated from the calibration curve and the dilution ratio.

## Neurological function and survival analysis

Neurological function was assessed 24 and 48 h after CA in surviving mice, using the published NFS [30]. Briefly, the NFS consists of six components: level of consciousness, corneal reflex, respiration, righting reflex, coordination, and movement/activity, with 0–2 points per component for a total score that ranges from 0 (no response or worst) to 12 (normal neurological function). The mice were monitored for survival until 7 days after CA. We predefined a humane endpoint as a 25% weight loss from the baseline weight. Mice were monitored and weighed daily until 7 days after CA. Once the humane endpoint was met, mice were euthanized immediately by cervical dislocation under anesthesia induced by the intraperitoneal administration of midazolam, medetomidine, and butorphanol. If any mice died before meeting the humane endpoint, this was documented, and humane endpoint criteria and monitoring procedures were reviewed and refined as necessary to prevent recurrence.

## Histological analysis

At 24 and 48 h post-CA, the mice were perfusion fixed, and their brains were excised. Coronal sections of 25-µm thickness were cut on a freezing microtome at the level of the hippocampus. To assess neuronal injury, brain slices were stained with FJC, labeling degenerative neuronal cells [31]. The percentages of FJC-positive areas were determined using an all-in-one fluorescence microscope (BZ-X800, KEYENCE, Osaka, Japan) and a hybrid cell count application (BZ-H4C, KEYENCE) in the BZ-X Analyzer software (BZ-H4A, KEYENCE) [32].

## Brain RNA analysis

Recent studies have reported that NMN attenuates neuronal injury via restoration of the SIRT3 pathway [12,33]. Additionally, peroxisome proliferator-activated receptor gamma coactivator 1-alpha (PGC-1α) regulates SIRT3 expression and preserves mitochondrial function [34]. Moreover, SIRT3 and PGC-1α reduce downstream IL-6 levels [25,35,36]. Elevated

IL-6 levels correlate with worse PCABI outcomes [37]. Based on these findings, this study investigated whether NAD$^+$ supplementation using NMN activated SIRT3 and PGC-1α and led to reduced IL-6 levels.

At 6 h post-ROSC, the hippocampi were dissected and snap frozen with liquid nitrogen. Brain RNA was isolated using an miRNeasy Mini Kit (Qiagen, Valencia, CA, USA) and converted to cDNA using the high-capacity reverse transcription kit (Applied Biosystems, Carlsbad, CA, USA), following the manufacturer's protocol. Quantitative polymerase chain reaction (qPCR) was performed using a QuantiFast SYBR Green PCR Kit (Qiagen). The qPCR primers used were as follows: *Sirt3* forward, 5′-ATCCCGGACTTCAGATCCCC-3′ and reverse, 5′-CAACATGAAAAAGGGCTTGGG-3′; peroxisome proliferative activated receptor gamma coactivator 1 alpha (*Ppargc1a*) forward, 5′-TATGGAGTGACATAGAGTGTGCT-3′ and reverse, 5′-CCACTTCAATCCACCCAGAAAG-3′; *Il6* forward, 5′-TAGTCCTTCCTACCCCAATTTCC-3′ and reverse, 5′-TTGGTCCTTAGCCACTCCTTC-3′; and glyceraldehyde-3-phosphate dehydrogenase (*Gapdh*) forward, 5′-GAAGGTCGGTGTGAACGGAT-3′ and reverse, 5′-ACTGTGCCGTTGAATTTGCC-3′ [38–40]. Relative RNA expression levels were calculated using the delta-delta cycle threshold method, normalized to *Gapdh* as a housekeeping gene. The qPCR was performed by an investigator blinded to the group allocation.

### Brain ELISA analysis

The protein levels of SIRT3 and IL-6 in the brain were measured using ELISA kits (SIRT3: SEE913Mu, Cloud-Clone Corporation, Katy, TX, USA; IL-6: M6000B-1, R&D Systems, Minneapolis, MN, USA) according to the manufacturer's instructions. Briefly, the hippocampi were homogenized in the radio-immunoprecipitation assay buffer containing 50 mM Tris-HCl (pH 8.0), 150 mM sodium chloride, 0.5 w/v% sodium deoxycholate, 0.1 w/v% sodium dodecyl sulfate, and 1.0 w/v% NP-40 substitute (RIPA Buffer, FUJIFILM, Osaka, Japan) with a protease inhibitor cocktail containing 1 mM AEBSF, 800 nM aprotinin, 50 μM bestatin, 15 μM E64, 20 μM leupeptin, and 10 μM pepstatin A (Halt™ Protease Inhibitor Cocktail, Thermo Fisher Scientific, Rockford, IL, USA). Homogenates were centrifuged at 10,000 × g for 5 min at 4°C, and the supernatant was then collected. Bicinchoninic acid assay of the supernatant was performed to quantify total protein concentrations (Pierce™ BCA Protein Assay Kits, Thermo Fisher Scientific). The levels of SIRT3 and IL-6 in the supernatant were measured using ELISA kits and normalized to the total protein concentration of the sample.

### Statistical analysis

The experimental results were presented as means with standard errors, medians with interquartile ranges (IQR), or numbers with percentages. Normally distributed variables were analyzed using two-tailed Student's t-tests, whereas non-normally distributed variables were analyzed using the Mann–Whitney U test. Categorical variables were analyzed using Fisher's exact test. One-way analysis of variance (ANOVA) was used to compare three or more groups. If the ANOVA test showed statistically significant differences, a post hoc two-tailed Student's t-test was performed. Survival was analyzed using a log-rank test by plotting the Kaplan–Meier curve. The prespecified statistical analysis of NFS at 24 and 48 h post-CA was performed among surviving mice only, with dead mice excluded from the comparisons in accordance with prior CA model studies [21,28,30]. Statistical significance was defined as a two-sided p-value of < 0.05. All statistical analyses were performed using Prism 10.3.0 (GraphPad Software Inc., San Diego, CA, USA).

### Results

#### Post-ROSC systemic administration of NMN increases NAD$^+$ and ATP concentrations in the brain

A total of 34 mice were used in this experiment. Four mice were used for the sham operation. A total of 2 mice did not achieve ROSC, and 28 mice were assigned to either the NMN or control group at 15 min and 2 h post-ROSC (n = 7 mice per group). The NAD$^+$ concentration in the brain decreased gradually after ROSC and was significantly lower 2 h

post-ROSC in control mice than in sham mice (sham: 169.7 ± 4.3 vs. control [2 h post-ROSC]: 91.0 ± 8.7 pmol/mg tissue; difference: −78.7 [95% confidence interval (CI): −106.2 to −51.2], p = 0.0001; Fig 2A). Systemic NMN administration successfully increased the NAD$^+$ concentration in the brain at 15 min post-ROSC (control: 141.4 ± 11.3 vs. NMN: 185.6 ± 14.8 pmol/mg tissue; difference: 44.2 [95% CI, 3.7 to 84.6], p = 0.04) and 2 h post-ROSC (control: 91.0 ± 8.7 vs. NMN: 231.2 ± 11.8 pmol/mg tissue; difference: 140.2 [95% CI, 108.3 to 172.0], p < 0.0001; Fig 2A). We measured brain ATP levels 2 h post-ROSC to assess the mitochondrial function in 10 mice assigned to either the NMN or control group (5 mice per group). NMN significantly increased the ATP levels in the brain 2 h post-ROSC (control: 203.6 ± 28.4 vs. NMN: 287.0 ± 21.9 nmol/g tissue; difference: 83.4 [95% CI, 0.8 to 166.0], p = 0.048; Fig 2B).

### NMN improves neurological function and survival after CA

Subsequently, we investigated whether the systemic administration of NMN improved post-CA outcomes. Among the 39 mice used in this experiment, 3 mice did not achieve ROSC, and thus, 36 mice were equally assigned to either the NMN or control group (n = 18 per group). The baseline characteristics, CPR time, and epinephrine dose were similar between the two groups (Table 1). There were also no significant differences in vital signs at 30 min after ROSC between groups. The baseline characteristics, resuscitation details, and vital signs of individual mice in experiment 2 are shown in S1 Table.

At 24 h post-CA, surviving mice in the NMN group showed a trend toward improved NFS (control: 7 [IQR: 6–11] vs. NMN: 12 [IQR: 5–12], p = 0.35; Fig 3A). Notably, NMN administration increased the number of mice without any neurological sequelae (an NFS of 12) at 24 h post-CA (control: n = 3 [17%] vs. NMN: n = 8 [44%], p = 0.14; Fig 3A). Furthermore, repeated NMN administration significantly improved NFS at 48 h post-CA in surviving mice (control: 8 [IQR: 4–12] vs. NMN: 12 [IQR: 9–12], p = 0.03; Fig 3A). The 7-day survival rate was significantly higher in the NMN group than in the control group (control: 22.2% [4/18] vs. NMN: 61.1% [11/18], p = 0.03; Fig 3B). No mice were sacrificed due to the humane endpoint. The NFS and survival times of individual mice are shown in S2 Table.

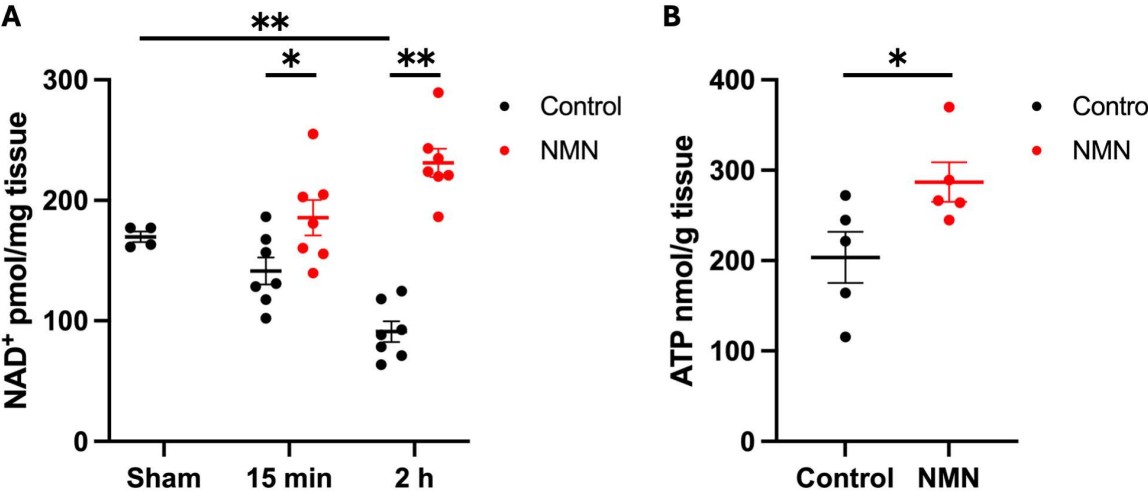

**Fig 2. Post-ROSC administration of NMN increases NAD$^+$ and ATP concentrations in the mouse brain.** (A) Brain NAD$^+$ concentration decreased after ROSC, and systemic NMN administration increased brain NAD$^+$ concentration at 15 min and 2 h post-ROSC. (B) NMN increased brain ATP concentration at 2 h post-ROSC. Data are presented as means with standard errors. (A) One-way ANOVA with a post hoc Student's t-test; (B) Student's t-test. *p < 0.05, **p < 0.01. ANOVA, analysis of variance; ATP, adenosine triphosphate; NAD$^+$, nicotinamide adenine dinucleotide; NMN, nicotinamide mononucleotide; ROSC, return of spontaneous circulation.

**Table 1. Comparison of baseline characteristics, resuscitation details, and vital signs between the two groups of mice.**

| | Control group (n=18) | NMN group (n=18) | p value[a] |
|---|---|---|---|
| Age (w) | 12.6±0.3 | 12.4±0.2 | 0.77 |
| Weight (g) | 25.5±0.4 | 25.7±0.4 | 0.76 |
| CPR time (min) | 1.8±0.1 | 2.0±0.1 | 0.12 |
| Epinephrine dose (µg) | 7.1 (6.0–8.1) | 6.3 (6.0–7.7) | 0.57 |
| **Physiological variables** | | | |
| **Baseline** | | | |
| HR (/min) | 386 (346–445) | 404 (357–423) | 0.72 |
| MAP (mmHg) | 88 (82–92) | 92 (88–97) | 0.78 |
| BT (°C) | 35.9±0.2 | 35.5±0.2 | 0.15 |
| **30 min after ROSC** | | | |
| HR (/min) | 604 (582–622) | 596 (567–601) | 0.27 |
| MAP (mmHg) | 105 (102–118) | 113 (110–116) | 0.13 |
| BT (°C) | 36.3±0.1 | 36.1±0.1 | 0.22 |

Data are presented as means with standard errors or medians with interquartile ranges.

[a]Student's t-test or Mann–Whitney U test.

BT, body temperature; CPR, cardiopulmonary resuscitation; CPR time, time from the start of CPR to ROSC; HR, heart rate; MAP, mean arterial pressure; NMN, nicotinamide mononucleotide; ROSC, return of spontaneous circulation.

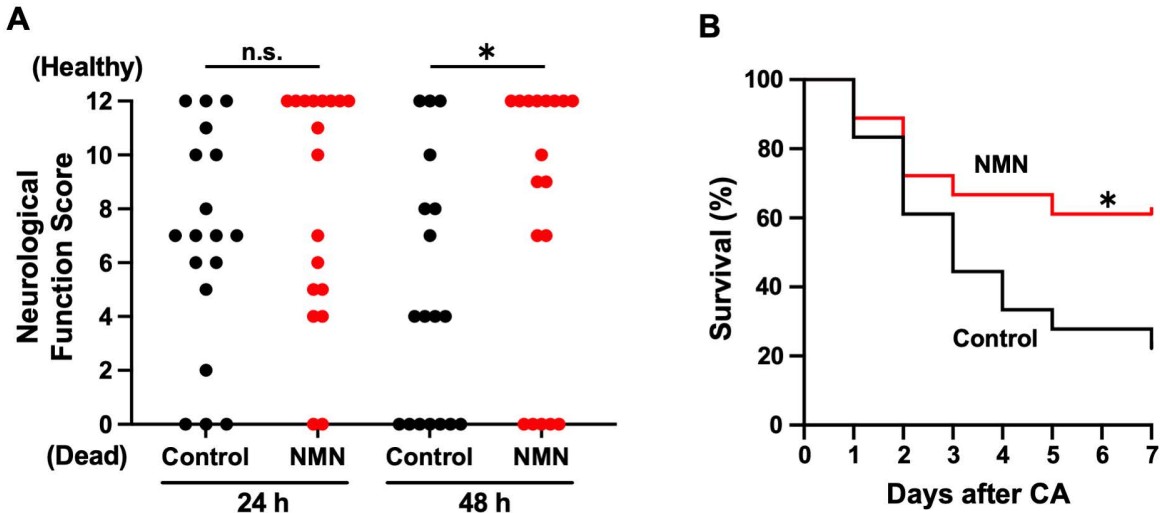

**Fig 3. NMN improves neurological function and survival after cardiac arrest in mice.** (A) Neurological function scores (NFS) at 24 and 48 h after cardiac arrest (n=18 per group; NFS analyzed only in surviving mice). (B) Seven-day survival after cardiac arrest (n=18 per group). (A) Mann–Whitney U test; (B) log-rank test. *p<0.05. CA, cardiac arrest; NFS, neurological function score; NMN, nicotinamide mononucleotide; n.s., not significant.

## NMN attenuates neuronal injury after CA

We evaluated hippocampal neuronal injury using FJC staining at 24 and 48 h post-CA, based on previous reports [22,41]. Four mice per group were used at 24 h post-CA, and 5 mice per group were used at 48 h post-CA. Hippocampal neuronal injury 24 h post-CA was found in control mice (Fig 4A). In accordance with the NFS findings, the NMN group showed a trend toward fewer FJC-positive areas following single NMN administration after ROSC (control: 1.4±0.2% vs. NMN: 0.9±0.3%; difference: −0.5 [95% CI, −1.3 to 0.3], p=0.16; Fig 4B). Neuronal injury progressed from 24 to 48 h post-CA in

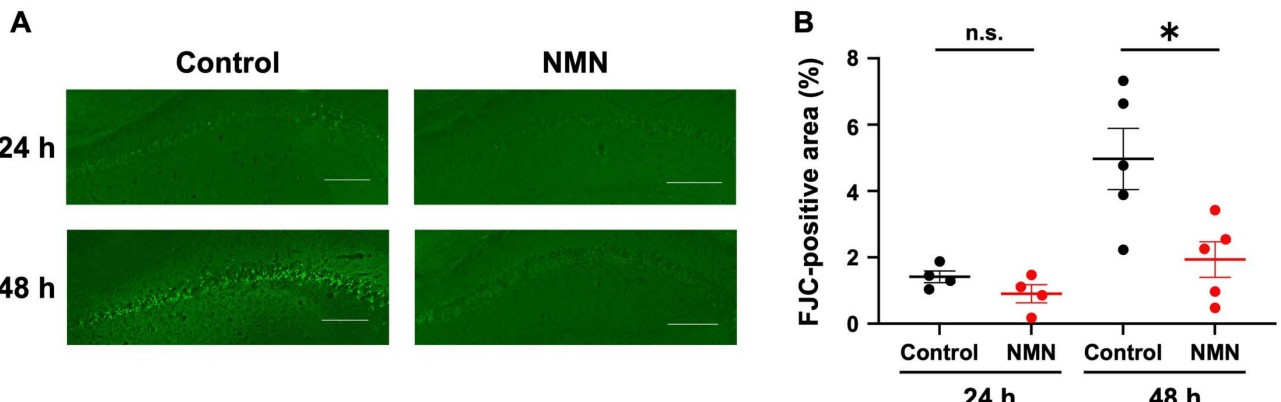

**Fig 4. NMN attenuates neuronal injury after cardiac arrest in mice.** (A) Representative histological images of hippocampi with FJC staining 24 and 48 h after cardiac arrest. Scale bar = 200 μm. (B) Quantitative assessment of FJC-positive areas of hippocampi 24 and 48 h after cardiac arrest. Data are presented as means with standard errors. (B) Student's t-test; *p < 0.05. FJC, Fluoro-Jade C; NMN, nicotinamide mononucleotide; n.s., not significant.

the control group (Fig 4A). In contrast, NMN attenuated the delayed neuronal injury at 48 h post-CA (Fig 4A), as evidenced by a significant reduction in FJC-positive areas (control: 5.0 ± 0.9% vs. NMN: 1.9 ± 0.5%; difference: −3.1 [95% CI, −5.5 to −0.6], p = 0.02; Fig 4B). The percentages of FJC-positive areas for individual mice are shown in S3 Table.

### NMN increases SIRT3 expression in the hippocampus after CA

We measured the mRNA expression of *Sirt3*, *Il6*, and *Ppargc1a* (the murine gene encoding PGC-1α, which regulates SIRT3 expression) (Fig 5A–C) and the protein levels of SIRT3 and IL-6 (Figs 5D and 5E) in the hippocampi of the control and NMN groups at 6 h post-CA. NMN significantly upregulated relative *Sirt3* expression (control: 1.0 ± 0.1 vs. NMN: 1.5 ± 0.1; difference: 0.5 [95% CI, 0.04 to 1.0], p = 0.04; Fig 5A) and increased SIRT3 protein levels (control: 17.7 ± 3.6 vs. NMN: 34.5 ± 4.4 pg/mg protein; difference: 16.8 [95% CI, 4.1 to 29.6], p = 0.01; Fig 5D). Moreover, NMN significantly downregulated relative *Il6* expression (control: 1.0 ± 0.2 vs. NMN: 0.4 ± 0.1; difference: −0.6 [95% CI, −1.1 to −0.03], p = 0.04; Fig 5B). IL-6 protein levels in the brain showed a trend to decrease in the NMN group, but the difference did not reach significance (control: 0.95 pg/mg protein [IQR: 0.73–1.37] vs. NMN: 0.56 pg/mg protein [IQR: 0.46–1.28], p = 0.24; Fig 5E). The NMN group also showed a trend toward higher *Ppargc1a* expression, although the difference did not reach significance (control: 1.0 ± 0.1 vs. NMN: 1.3 ± 0.1; difference: 0.3 [95% CI, −0.2 to 0.7], p = 0.15; Fig 5C). The qPCR cycle threshold values and specific protein levels of individual mice are shown in S4 Table. In summary, NMN increased SIRT3 expression in the hippocampus early after CA.

## Discussion

This study revealed that systemic NMN administration post-ROSC increased NAD⁺ and ATP levels in the brain, improved neurological function and survival, and attenuated histological neuronal injury in a murine model of CA. This concurrently occurred with SIRT3 activation in the brain. To the best of our knowledge, this is the first report on the therapeutic potential of post-CA systemic administration of NMN for mitigating PCABI.

Brain NAD⁺ levels decreased post-ROSC, and intraperitoneal NMN administration increased brain NAD⁺ levels 15 min and 2 h post-ROSC, consistent with findings from a previous study on wild-type mice [11]. The rapid replenishment of NAD⁺ concentrations in the brain after systemic NMN administration is particularly promising for clinical translation, suggesting systemic NMN administration as an adjunct therapy in post-arrest critical care.

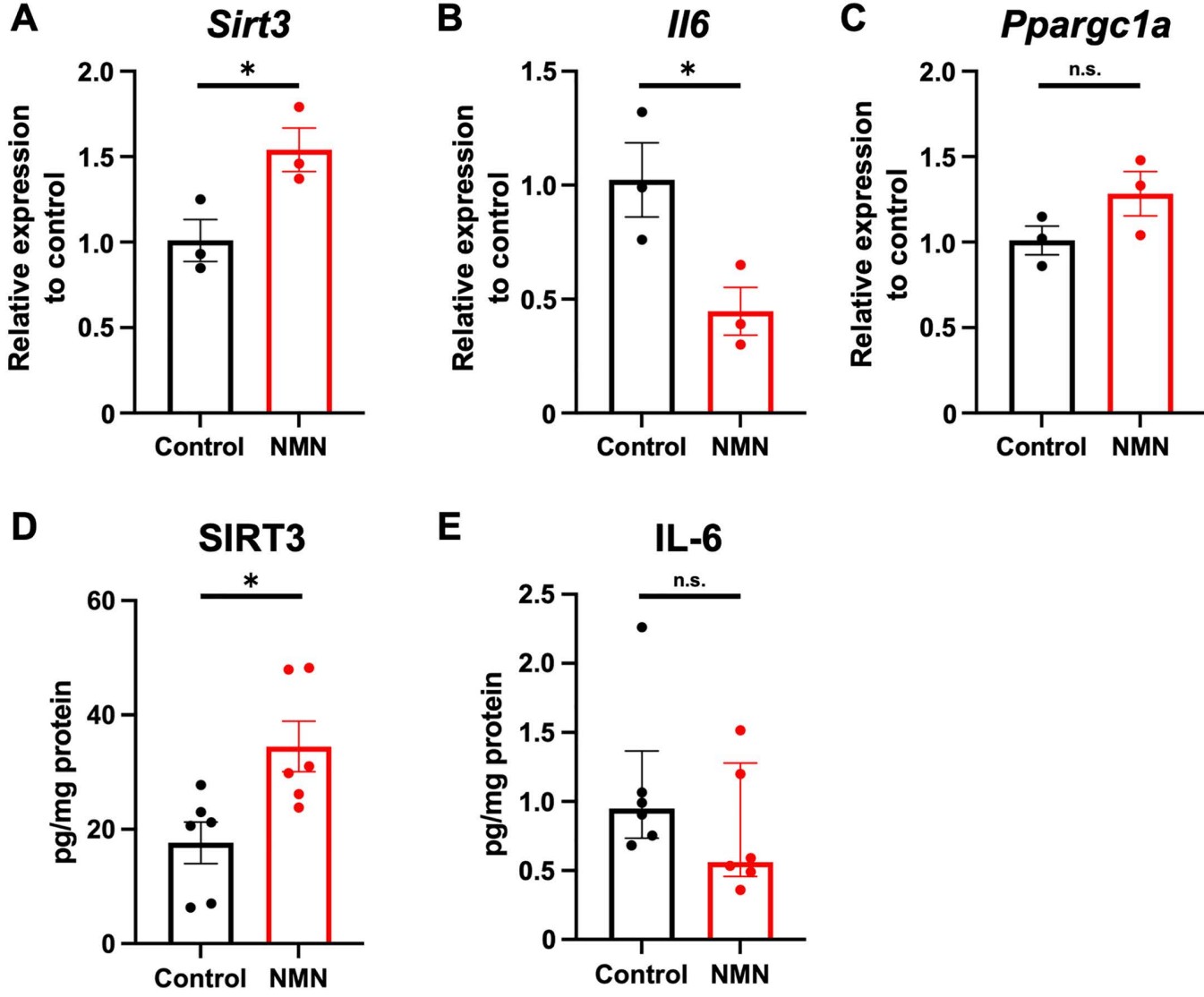

**Fig 5. NMN increases SIRT3 expression at both mRNA and protein levels in the brain 6 h after cardiac arrest.** Relative mRNA expression in hippocampi 6 h after cardiac arrest (n = 3 per group): (A) *Sirt3*, (B) *Il6*, and (C) *Ppargc1a*. Hippocampal protein levels of SIRT3 and IL-6 6 h after cardiac arrest (n = 6 per group): (D) SIRT3, and (E) IL-6. Data are presented as means with standard errors or medians with interquartile ranges. (A–D) Student's t-test; (E) Mann–Whitney U test. *p < 0.05. *Il6*, interleukin-6 (gene); IL-6, interleukin-6 (protein); NMN, nicotinamide mononucleotide; n.s., not significant; *Ppargc1a*, peroxisome proliferator-activated receptor gamma coactivator 1-alpha; RNA, ribonucleic acid; *Sirt3*, sirtuin-3 (gene); SIRT3, sirtuin-3 (protein).

A relatively low dose of 60 mg/kg NMN was selected in our study based on a previous study showing the greatest neuroprotective effects at this low dose in a murine model of cerebral ischemia [10]. A recent study suggested that low doses of NMN may help prevent the undesired effects associated with excessive NMN, which can aggravate neurodegeneration by activating sterile alpha and toll/interleukin-1 receptor motif-containing 1 (SARM1) [42]. Additionally, low-dose NMN was administered 3 times (0, 24, and 48 h) based on previous findings [3,10]. Future studies will address the effect of different NMN administration protocols against PCABI.

NMN administration improved neurological function at 48 h post-CA among surviving mice (Fig 3A). In this study, we analyzed NFS only in surviving mice, as this approach more accurately reflects the neurological outcomes without

conflating neurological impairment with mortality. This approach of evaluating NFS exclusively in surviving mice after CA has been validated in multiple studies and has become the prevailing standard in recent years [21,28,30]. The original publication of the mouse CA model assessed NFS by including deceased mice [43]. However, in that study, a large number of mice had died by 18 and 72 h post-CA, and no difference in NFS was observed when analyzing only the surviving mice. Although acute mortality is often attributed to PCABI, deaths from cardiovascular instability or multiorgan failure can also occur [1]. In animal models, the exact cause of death is often unclear even upon autopsy, and not all deaths can be attributed to brain injury. In our experiments, no mice were sacrificed due to the humane endpoint related to brain injury. Therefore, including deceased mice in the analysis of neurological outcomes with an NFS of 0 could introduce bias, as deaths from non-neurological causes would also be classified as poor neurological outcomes.

NMN administration also increased the 7-day survival rate. Notably, survival between the control and NMN groups began to differ after 72 h post-CA, when PCABI becomes the primary cause of death [1] (Fig 3B). Considering that differences in neurological function were already evident before 72 h post-CA (Fig 3A), it is plausible that neurological dysfunction contributes to mortality in this study. The cardiovascular effects of NMN were unclear in our study. There was no significant difference in MAP between the two groups at 30 min post-ROSC (Table 1). Similarly, the cardiovascular effects of NMN on other critical illnesses remain controversial [10,44]. Low-dose NMN (62.5 mg/kg) did not improve cerebral blood flow in a murine model of cerebral infarction [10]. In contrast, high-dose NMN (500 mg/kg) improved left ventricular systolic function in a murine model of myocardial ischemia [44]. Whether the cardiovascular effects of NMN are dose dependent or only due to the differences between focal or global ischemia-reperfusion requires further investigation.

Neuronal injury progressed from 24 h to 48 h post-CA, consistent with a previous study reporting delayed neuronal injury within 2–3 days post-CA [45]. Notably, at 48 h post-CA, neuronal injury was more significantly mitigated in the NMN group than in the control group, suggesting that NMN alleviated the progression to delayed neuronal injury.

NMN also significantly increased ATP levels in the brain post-CA (Fig 2B). ATP levels may reflect mitochondrial function because mitochondria are the main organelles responsible for ATP production [46]. Cellular ATP levels can decrease after ischemia-reperfusion due to a delayed perturbation of mitochondrial function [47]. Therefore, the increased ATP levels in the NMN group may suggest improved mitochondrial function. Moreover, we found that NMN administration increased post-CA SIRT3 levels in the brain (Fig 5). NMN has been reported to preserve post-ischemic mitochondrial function via SIRT3-dependent mechanisms, including mitochondrial protein deacetylation, normalization of ROS production, and inhibition of mitochondrial fragmentation [12]. Our findings collectively suggest that NMN possibly improves post-CA mitochondrial function via SIRT3 activation. The direct mechanisms through which NMN improves mitochondrial function warrant further mechanistic studies. PGC-1α is a marker of mitochondrial biogenesis and interacts with SIRT3. The current study measured the expression of *Ppargc1a* in the brain post-CA and found no significant difference between groups (Fig 5C). This may be attributable to SIRT3 being regulated by multiple pathways in addition to PGC-1α.

NMN exerts anti-inflammatory effects, including the attenuation of IL-6 production in models of intracerebral hemorrhage, traumatic brain injury, and sepsis-associated encephalopathy [17,25,48]. Increased systemic IL-6 levels are associated with increased mortality and poor neurological prognosis following CA [49–51]. Therefore, we next investigated whether NMN improved post-CA neuronal injury through the attenuation of IL-6 levels. The NMN group showed significantly lower *Il6* expression post-CA (Fig 5B), whereas there was no significant difference in the protein levels of IL-6 between groups (Fig 5E). This may be due to the suboptimal timing of protein measurement for detecting differences, as protein synthesis from mRNA requires time.

This study had limitations. First, the study was single-blinded. Although the investigators who performed the experiments knew the treatment assignments, those who assessed neurological function and histological neuronal injury were blinded to the treatment. Second, this is a proof-of-principle study, and the mechanisms by which NMN improves delayed neuronal injury after CA have not yet been fully elucidated. Mechanisms involving sirtuins other than SIRT3, particularly in mitochondrial function and anti-inflammation, should be elucidated in the future [2,4,52]. Third, the long-term

hemodynamic effects of NMN after CA are not understood. Fourth, the protocol for NMN administration after CA (e.g., route [especially intravenously], timing, and dosage) needs to be optimized. For future clinical translation, delayed NMN administration should be considered, as neuronal cells are protected by NMN administration even at 30 min after cerebral ischemia-reperfusion in the model of cerebral infarction [10]. Finally, although the beneficial effects of NMN on aging have been reported [4], this study used young mice. As CA occurs more frequently in the elderly population, future studies should clarify the effects of NMN in post-CA elderly mice. Despite these limitations, this proof-of-principle study presents robust evidence of NMN as a promising therapeutic option for PCABI.

## Conclusions

Systemic NMN administration post-CA increased brain NAD$^+$ and ATP levels, attenuated delayed neuronal injury 48 h post-CA, and improved neurological function and survival in a murine model of CA. These concurrently occurred with SIRT3 activation in the brain. Systemic NMN administration is a promising therapeutic approach for replenishing NAD$^+$ in the brain and mitigating PCABI.

## Supporting information

**S1 Table. Baseline characteristics, resuscitation details, and vital signs of individual mice in experiment 2.**
(DOCX)

**S2 Table. Neurological function score and survival time of individual mice in experiment 2.**
(DOCX)

**S3 Table. Percentages of FJC-positive areas for individual mice in experiment 3.**
(DOCX)

**S4 Table. qPCR cycle threshold values and protein levels of individual mice in experiment 4.**
(DOCX)

## Author contributions

**Conceptualization:** Tomoyoshi Tamura, Jun Yoshino, Koichiro Homma.

**Data curation:** Daiki Kaito, Tomoyoshi Tamura, Sayuri Suzuki.

**Formal analysis:** Daiki Kaito, Tomoyoshi Tamura, Tadashi Matsuoka, Koichiro Homma.

**Funding acquisition:** Tomoyoshi Tamura, Jun Yoshino, Koichiro Homma.

**Investigation:** Daiki Kaito, Tomoyoshi Tamura, Sayuri Suzuki, Ryutaro Onishi.

**Methodology:** Daiki Kaito, Tomoyoshi Tamura, Sayuri Suzuki, Kenji F. Tanaka, Tadashi Matsuoka, Katsuya Maeshima, Koichiro Homma.

**Project administration:** Daiki Kaito, Tomoyoshi Tamura, Sayuri Suzuki, Tadashi Matsuoka, Katsuya Maeshima, Junichi Sasaki, Koichiro Homma.

**Resources:** Tomoyoshi Tamura, Sayuri Suzuki, Kenji F. Tanaka, Jun Yoshino, Junichi Sasaki, Koichiro Homma.

**Supervision:** Junichi Sasaki, Koichiro Homma.

**Visualization:** Daiki Kaito, Tomoyoshi Tamura, Ryutaro Onishi, Kenji F. Tanaka.

**Writing – original draft:** Daiki Kaito.

**Writing – review & editing:** Daiki Kaito, Tomoyoshi Tamura, Sayuri Suzuki, Ryutaro Onishi, Kenji F. Tanaka, Jun Yoshino, Tadashi Matsuoka, Katsuya Maeshima, Junichi Sasaki, Koichiro Homma.

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
