## [Decision Letter · Decision Letter 0]

11 Feb 2025

Dear Dr. Homma,

Thank you for submitting your manuscript to PLOS ONE. After careful consideration, we feel that it has merit but does not fully meet PLOS ONE’s publication criteria as it currently stands. Therefore, we invite you to submit a revised version of the manuscript that addresses the points raised during the review process.

We look forward to receiving your revised manuscript.

Kind regards,

Jean Baptiste Lascarrou

Academic Editor

PLOS ONE

Journal Requirements:

2. Thank you for your submission to PLOS ONE. We note that your study design may include death of a regulated animal as a likely outcome or planned experimental endpoint. At this time, we request that you please report additional details in your Methods section regarding animal care and use for the survival study, as per our editorial guidelines (http://journals.plos.org/plosone/s/submission-guidelines#loc-humane-endpoints ).

For easy reference, we have attached a checklist that may be relevant for your submission. Please complete all items on the checklist at the following link: http://journals.plos.org/plosone/s/file?id=bb1d/plos-one-humane-endpoints-checklist.docx

Please upload the completed checklist as file type “Other” when resubmitting your manuscript. This document is for internal journal use only and will not be published if your article is accepted. We very much appreciate your attention to these requests and support of improved reporting standards in PLOS ONE submissions.

3. To comply with PLOS ONE submissions requirements, in your Methods section, please provide additional information regarding the experiments involving animals and ensure you have included details on (1) methods of sacrifice and (2) efforts to alleviate suffering.

[I have read the journal's policy and the authors of this manuscript have the following competing interests: [J.Y. is listed as an inventor on patent applications related to NMN and adiponectin (US20180228824, JP2018131418A). The other authors declare no conflict of interest.].

Please confirm that this does not alter your adherence to all PLOS ONE policies on sharing data and materials, by including the following statement: ""This does not alter our adherence to PLOS ONE policies on sharing data and materials.” (as detailed online in our guide for authors http://journals.plos.org/plosone/s/competing-interests ). If there are restrictions on sharing of data and/or materials, please state these. Please note that we cannot proceed with consideration of your article until this information has been declared.

Reviewers' comments:

Reviewer's Responses to Questions

**Comments to the Author**

1. Is the manuscript technically sound, and do the data support the conclusions?

Reviewer #1: Partly

Reviewer #2: Yes

2. Has the statistical analysis been performed appropriately and rigorously?

Reviewer #1: Yes

Reviewer #2: Yes

3. Have the authors made all data underlying the findings in their manuscript fully available?

Reviewer #1: No

Reviewer #2: Yes

4. Is the manuscript presented in an intelligible fashion and written in standard English?

Reviewer #1: Yes

Reviewer #2: Yes

Reviewer #1: I read with interest the manuscript entitled « Systemic nicotinamide mononucleotide administration to mitigate post-cardiac arrest brain injury in mice » by Koichiro Homma et al. The authors provided robust data suggesting that nicotinamide mononucleotide administered intraperitoneally (first dose upon restoration of circulation) could improve neurological outcomes after KCl-induced cardiac arrest in mice. This is a first step towards potential application in human. They also tried to provide mechanistic insights by performing qPCR that revealed an increase in Sirtuin 3 expression and a decrease in IL-6 expression in brain of treated animals. Even though data on survival and brain damage are quite convincing, opening new opportunities to treat humans in the future, the mechanistic explorations are weak. I have the following comments:  

Major  

- The conclusion of the abstract stating that NMN restored mitochondrial biogenesis is not supported by the results. The authors must provide data on biogenesis to support this claim. The conclusion of the main article stating that “Nicotinamide mononucleotide showed these favorable effects at least in part through the restoration of the Sirt3/Ppargc1a pathway and reduction of Il-6 expression in the brain. » is not sufficiently supported by the results. There is an association, but no demonstration, of the mechanisms of protection.  

- This study can be interpreted as a preclinical pilot study. The authors justified the choice of testing NMN, whereas the protective effects of NAD+ administration with niacin have been demonstrated   because it is more tolerated and efficient to enter the cell than other drugs that can increase NAD+ levels. In this case, why did the authors not choose a route of administration (IV) that could be used in patients with CA?  

- NMN was administered immediately after ROSC. As shown by all RCT, it is not possible to administer a new drug within minutes after ROSC, which may explain the lack of translation of many promising drugs in clinical trials. It would be interesting to test whether delaying the administration of NMN is still protective in an additional group (to ensure that it can be tested in clinical trials). In any case, this point should be discussed from the perspective of translational medicine.  

- For NAD levels (Experiment 1), can the authors provide data on sham animals? The question is whether NMN restores CA-induced depletion of NMN. Moreover, we tested whether NMN interfered with NAD + measurement. Finally, how did the authors ascertain that NAD+ was measured in the brain cells and not in the extracellular fluid? (NMN may have modified blood NAD+ level without increasing brain cells NAD+ levels)  

- It is unclear whether the investigators who performed surgery and CPR were blinded to the experimental group. We understand that this was not the case in the limitations paragraph of the Discussion section. Please revise the methods to make them clearer to the reader.  

- The dose of epinephrine administered during resuscitation has not been reported in the Results section. Therefore, it is important data to provide to ensure that the two groups of animals are comparable.  

- The authors measured SIRT3 expression as well as IL-6 and found an increase and a decrease, respectively. However, it is unclear why they did not measure SIRT3 levels and activity IL-6 levels. Moreover, the authors did not provide data directly linking NMN administration with the abovementioned results. Finally, the authors declared that “Nicotinamide mononucleotide improved Sirt3/Ppargc1a expression »  (Line 268) which is not completely true as Ppargc1a expression  was not significantly different between the 2 experimental groups.

Minor   

Introduction 

-Reference 3, published in 2013, is not really « recent » as suggested by the authors in the introduction (line 46, page 3) ;

- Please, clarify how CA was induced in the abstract as the KCl-induced CA is not a usual cause of CA - 

- Justify the intraperitoneal route for the adminstration of NMN (rather than intravenous)  

Results

- There are elements of methods or discussion in the Results: See, for example lines 172-174 (Mitochondrial injury occurs rapidly after ROSC [24], making timely transport of NMN across the blood– brain barrier crucial for preventing and treating PCABI. A previous study reported that NMN rapidly increases brain NAD+ levels [11]). Please revise the entire section accordingly.

- Line 180: these data are already available in Figure 2

- Line 193 : « Despite a longer CPR time in the NMN group, MAP showed an elevated trend in the NMN group at 30 min after ROSC. » Given the results in Table 1, there was no difference between the two groups regarding these parameters. Please revise.

- Please provide body temperature at baseline in Table 1  

Discussion:  

- Line 280 Nicotinamide mononucleotide should be changed in « NMN ». See also line 320.

-  Lines 293-296. See the last point on the major limitations. The same is true for the next paragraph.

Reviewer #2: Kaito et al. investigated the effects of nicotinamide mononucleotide (NMN) administration on neurological outcomes in a mouse model of cardiac arrest. NMN supplementation aims to replenish NAD+ levels, which become depleted following ischemia. The authors conducted a comprehensive study comprising four successive experimental approaches to assess cerebral NAD+ concentrations, neurological dysfunction and survival, brain lesions, and transcript levels of cerebral inflammatory markers. This is an elegant in vivo study, with well-designed experiments that provide valuable insights into the topic. The manuscript is well written, and the data are clearly presented. Notably, all individual data points are available in the supplementary materials, ensuring transparency and reproducibility.

I have only minor comments:

• Page 5, Line 90: "Sample size for the outcomes..." – Could the authors clarify the initial hypothesis regarding the expected difference? Specifically, please provide the anticipated effect size along with the expected mean and standard deviation (SD) under control conditions.

• Page 6, Line 120: The route, timing, dosage, and volume of NMN administration should be explicitly stated in this section rather than solely in the Results.

• Page 9, Lines 172–175: This paragraph should be moved to the Introduction of the Methods section, as it is not a result but rather methodological background.

• Figure 1: The timing of brain sample collection in Experiments 1 and 4 is unclear and should be explicitly indicated in the figure.

• Figure 3: The Methods section states that animals that died were excluded from the neurological dysfunction analysis. However, in the figure, several scores appear at 12. Could the authors clarify whether these scores correspond to surviving animals? Additionally, since many scoring systems use death as the maximum score, it would be helpful to specify how mortality is represented in the figure. I believe that 12 corresponds to dead animals and that these animals just did not participate to the statistical analysis but this is counfounding if mentionned in the figure. If it is right, the authors could invert left and right panel and presents only alive animal in the NDS panel.

**Do you want your identity to be public for this peer review?** For information about this choice, including consent withdrawal, please see our Privacy Policy

Reviewer #1: No

Reviewer #2: No

---

## [Author Response · Author response to Decision Letter 1]

25 Jun 2025

Thank you for the constructive suggestions to improve our original manuscript. Our manuscript has benefited from these insightful suggestions. A summary of the revisions is as follows:

Reviewer 1:

1. We revised mechanistic claims throughout the manuscript. To support the role of NMN in mitochondrial function, we added new brain ATP measurements and updated Figure 2 accordingly.

2. The use of intraperitoneal administration was justified based on previous literature and the constraints of our model. We also discussed the clinical relevance and limitations of intravenous administration.

3. We added discussion on the possibility of delayed NMN administration, referencing prior studies, to enhance clinical translatability.

4. We revised Figure 2 to include brain NAD⁺ data from the sham mice, and NMN and control mice at 2 hours after ROSC.

5. We clarified the blinding protocol in the Methods section.

6. Epinephrine dose data were added to Table 1, demonstrating no significant difference between groups.

7. We measured brain SIRT3 and IL-6 protein levels and revised Figure 5 and mechanistic claims.

8. We revised the manuscript throughout, including the Abstract, Results, and Discussion, to address reviewer feedback.

Reviewer 2:

1. We added detailed sample size calculations, including expected means, standard deviations, and effect sizes for each outcome measure.

2. The full NMN administration protocol was clearly described in the Methods section.

3. The paragraph containing the methodological background, which was originally placed in the Results section, has been moved to the Methods section.

4. Figure 1 was revised to indicate the exact timing of brain sample collection.

5. In Figure 3, we clarified that the neurological function score (NFS) ranges from 0 (dead) to 12 (healthy), and revised the figure panel to show only surviving animals, because dead animals were excluded from statistical analysis.

---

## [Decision Letter · Decision Letter 1]

26 Aug 2025

Dear Dr. Homma,

We look forward to receiving your revised manuscript.

Kind regards,

Jean Baptiste Lascarrou

Academic Editor

PLOS ONE

Journal Requirements:

Reviewers' comments:

Reviewer's Responses to Questions

**Comments to the Author**

Reviewer #1: (No Response)

Reviewer #2: All comments have been addressed

2. Is the manuscript technically sound, and do the data support the conclusions?

Reviewer #1: Partly

Reviewer #2: Yes

3. Has the statistical analysis been performed appropriately and rigorously?

Reviewer #1: No

Reviewer #2: Yes

4. Have the authors made all data underlying the findings in their manuscript fully available?

Reviewer #1: Yes

Reviewer #2: Yes

5. Is the manuscript presented in an intelligible fashion and written in standard English?

Reviewer #1: Yes

Reviewer #2: Yes

Reviewer #1: R2 - Systemic nicotinamide mononucleotide administration to mitigate post-cardiac arrest

brain injury in mice » by Koichiro Homma et al.

I thank the authors for their thorough responses to my previous comments and for the significant

improvements made to the manuscript. The revisions have strengthened the study.

Nevertheless, I still have one substantive concern regarding the analysis of the neurological

function score, which is central to the manuscript’s main conclusion.

In response to reviewer 2, the authors modified Figure 3. However, in Figure 3A, neurological

scores are reported as significantly higher at 48 hours post-ROSC, while it seems that animals

who died were excluded from this analysis. This approach is problematic, since the scoring

system assigns a value of 0 to deceased animals. When recalculating the statistics using the

individual scores provided in Table S2, and including the deceased animals (as described in the

original method by Neumar et al., Circulation 2004; 109:2786–2791), the difference in

neurological outcome is no longer statistically significant (p = 0.08).

Therefore, the manuscript’s principal conclusion—that nicotinamide improves neurological

function—requires revision. I recommend that the authors present the analysis transparently,

including all animals in accordance with the validated scoring system, and adjust both the

results and their interpretation accordingly. If the authors choose to retain the current analysis,

they should provide a clear justification and cite supporting validation from the literature, and

describe the causes of death of the excluded animals as well as the humane endpoints if death

did not occur spontaneously

Reviewer #2: Thanks for the revision. No additional comment.

**Do you want your identity to be public for this peer review?** For information about this choice, including consent withdrawal, please see our Privacy Policy

Reviewer #1: No

Reviewer #2: **Yes: ** Renaud Tissier

---

## [Author Response · Author response to Decision Letter 2]

16 Sep 2025

We sincerely appreciate the thoughtful comments from the reviewers and the editor. We have carefully rechecked and revised the manuscript. The main points addressed in this revision are summarized below:

1. Based on biological rationale and previous publications, we chose to analyze the neurological function score (NFS) only among surviving mice and to report survival separately.

2. In response to Reviewer #1’s point regarding transparent data communication, we added the deceased mice to the figure panel (Fig 3A).

3. We revised the manuscript to emphasize that NFS analyses were performed solely on surviving mice.

---

## [Editor Report · Decision Letter 2]

30 Sep 2025

Systemic nicotinamide mononucleotide administration to mitigate post-cardiac arrest brain injury in mice

PONE-D-24-59975R2

Dear Dr. Homma,

We’re pleased to inform you that your manuscript has been judged scientifically suitable for publication and will be formally accepted for publication once it meets all outstanding technical requirements.

Kind regards,

Jean Baptiste Lascarrou

Academic Editor

PLOS ONE
---

## [Editor Report · Acceptance letter]

PONE-D-24-59975R2

PLOS ONE

Dear Dr. Homma,

I'm pleased to inform you that your manuscript has been deemed suitable for publication in PLOS ONE. Congratulations! Your manuscript is now being handed over to our production team.

Kind regards,

on behalf of

Dr. Jean Baptiste Lascarrou

Academic Editor

PLOS ONE